# Possible Third Step Preventing Conjugation between Different Species of *Blepharisma*

**DOI:** 10.3390/microorganisms11010188

**Published:** 2023-01-12

**Authors:** Ayu Sugino, Mayumi Kobayashi, Mayumi Sugiura, Terue Harumoto

**Affiliations:** 1Biological Sciences Course, Graduate School of Humanities and Sciences, Nara Women’s University, Nara 630-8506, Japan; 2Division of Science, Graduate School of Humanities and Sciences, Nara Women’s University, Nara 630-8506, Japan; 3Research Group of Biological Sciences, Division of Natural Sciences, Nara Women’s University, Nara 630-8506, Japan; 4Center for Diversity and Inclusion, Nara Women’s University, Nara 630-8506, Japan

**Keywords:** ciliate, interspecific conjugation, mating pheromone, pair formation, meiosis, macronuclear anlagen, progeny, reproductive isolation

## Abstract

In the genus *Blepharisma*, reproductive isolation between different species appears to be established at least by two barriers: (1) a mating pheromone, i.e., gamone 1, and (2) a factor involved in pair formation. Using four species, we experimentally investigated other potential barriers to interspecific conjugation in *Blepharisma,* as well as the first and second barriers. Cell-free fluid from type I cells (CFF1) of *B. americanum* had no effect on *B. undulans*, *B. japonicum*, or *B. stoltei*. Type II cells of *B. americanum* responded to CFF1 from *B. americanum* but not to CFF1 from *B. undulans*, *B. japonicum*, or *B. stoltei*. Gamone 1, therefore, seems to be the first reproductive barrier (with the inclusion of *B. americanum* species [megakaryotype 3]) as reported previously. In pretreated cells with complementary gamones in *B. undulans* and *B. americanum*, inter-species pair formation was rare, but pair formation between *B. americanum* and *B. japonicum* and between *B. americanum* and *B. stoltei* occurred at relatively high frequency. Most of the inter-species *B. americanum*–*B. stoltei* pairs underwent nuclear changes specific to conjugation. No significant difference was observed between the intra- and inter-species pairs over the time course of the nuclear changes, but the percentage of abnormal cells was higher in inter-species pairs than in intra-species pairs, and no progenies were produced by inter-species pairs. These results suggest a third barrier or step, in addition to the first and second ones, in nuclear changes after pair formation that prevents interspecific conjugation in *Blepharisma*.

## 1. Introduction

Conjugation is a mode of sexual reproduction in ciliates that usually occurs within the same species; interspecific conjugation is relatively uncommon among ciliates. However, interspecific mating has been observed between *Euplotes octocarinatus* and *E. patella*, *Paramecium caudatum* and *P. tetraurelia*, *P. caudatum* and *P. multimicronucleatum*, and *P. tetraurelia* and *P. multimicronucleatum* [1,2]. Inter-species conjugation rarely occurs in the genus *Blepharisma*, and at least two barriers that block conjugation between different *Blepharisma* megakaryotypes (Mks) have been suggested [3].

In *Blepharisma* species, conjugation is usually initiated by the interaction between complementary mating-type cells (I and II) via mating pheromones, i.e., the so-called ‘gamones’ in the same species. Under food-deprived conditions, mating type I cells secrete the mating pheromone gamone 1, and mating type II cells secrete the mating pheromone gamone 2. These pheromones stimulate each other, and the cells then form mating pairs; the conjugation process thus takes place [4]. Gamone 1 is a glycoprotein with N-linked oligosaccharides [5,6] encoded by *gamone 1* gene [7]. Gamone 2 was identified as 3-(2′-formylamino-5′-hydroxybenzoyl) lactate [8]. Gamone 1 is species-specific as far as reported, and gamone 2 is common among *Blepharisma* species [9].

During the initiation of conjugation, gamones 1 and 2 induce at least four phenomena: (1) the promotion of the secretion of another gamone in complementary mating-type cells [5], (2) the attraction of complementary mating-type cells [10,11,12], and (3) the transformation of complementary mating-type cells, enabling the cells to acquire the competence of adhesion first at the cilia and then at the cell surface [13,14]. (4) the induction of mating-type specific gene expression [15]. After these phenomena, both homotypic pairs (consisting of the same mating-type cells) and heterotypic pairs (consisting of complementary mating-type cells) are formed. Only cells of heterotypic pairs undergo meiosis and exchange one of two gametic nuclei and then generate fertilized nuclei [16]. Meiosis does not occur in homotypic pairs, and the pairs eventually separate [15,17].

More than 20 morphological species have been reported in the genus *Blepharisma*, and they are classified into four megakaryotypes (Mks) on the basis of their macronuclear morphology [18]: megakaryotype I (Mk1), characterized by a compact macronucleus; megakaryotype II (Mk2), with a binodal macronucleus; megakaryotype III (Mk3), showing a multinodal macronucleus; and megakaryotype IV (Mk4), with a filiform macronucleus. Several morphological species are included in each Mk class. *B. americanum* and *B. musculus* are classified as Mk3; they secrete species-specific gamone 1 and have no effect on each other [9]. They are thus known as reproductively isolated species. In contrast, the Mk4-class species *B. japonicum* and *B. stoltei* have always been questioned [19]. These species secrete gamone 1, which is effective on both species and may produce viable progenies. *B. japonicum* and *B. stoltei* were now recognized as closely related species in light of their mutual effectiveness of gamone 1 and the amino acid sequence homology of gamone 1 [3].

It has been experimentally studied that the reproductive isolation between different species appears to be established by at least two barriers [3]. The first barrier concerns the mating pheromone, gamone 1, as described above. The second barrier appears to occur during the process of pair formation. When both type I and type II cells of two different species are present, gamone 1 is secreted from type I cells, with a specific effect on the type II cells of each species; the type II cells of both species are stimulated and secrete gamone 2, which universally affects type I cells of both species. Thus, all cells are stimulated and pair formation even between different species seems to be possible. However, this was not the case when *B. undulans* (Mk2) and *B. japonicum* (Mk4) were used [3]. Even when the cells were stimulated, an interspecific pairing rate <2% was observed. The second barrier to prevent interspecific conjugation thus appears to occur at the mating stage involving a possible factor for pair formation [3].

We conducted the present study to confirm previous study and further investigate potential barriers to inter-species conjugation in *Blepharisma*. We used newly obtained Mk3 stocks to determine whether artificially stimulated cells form inter-species pairs (*B. undulans* (Mk2) and *B. americanum* (Mk3), *B. americanum* (Mk3) and *B. japonicum* (Mk4), and *B. americanum* (Mk3) and *B. stoltei* (Mk4)). When such inter-species pair formation was successful, we also determined whether the further conjugation steps such as meiosis of micronuclei, the formation of pronuclei, and the fertilization and development of new macronuclear anlagen occur normally. We also sought to clarify whether such interspecific conjugation produces viable progenies. The possibility of additional barriers or steps preventing further conjugation between different *Blepharisma* species is discussed.

## 2. Materials and Methods

### 2.1. Strains and Culture

The stocks used in this study are listed in Table 1. The newly obtained stocks were CCAP1607/1 and KR-III. Stock CCAP1607/1 was purchased from the Culture Collection of Algae and Protozoa (CCAP, Dunbeg, Scotland). Stock ATCC30299 was purchased from the American Type Culture Collection (ATCC, Manassas, VA, USA). All stocks have been cultivated and maintained in our laboratory. Cells were cultured at 25 °C in diluted fresh lettuce juice medium inoculated with *Enterobacter aerogenes* 2 days before use. When the cells reached the stationary phase, they were concentrated by mild centrifugation, washed with SMB (Synthetic Medium for *Blepharisma*: physiologically balanced solution, SMB III medium without EDTA) [4,7] and suspended in SMB at 1500–2000 cells/mL density.

### 2.2. Preparation of Cell-Free Fluid (CFF)

To obtain CFF containing gamone 1, we incubated moderately starved type I cells at 25 °C for 2 days. The cells were then removed from the suspension by mild centrifugation, and the remaining CFF was filtered through a nylon net (mesh size 1 × 1 µm) and then through a 0.20 μm DISMIC-25 cs filter. The prepared CFF was designated as CFF1.

### 2.3. Preparation of Synthetic Gamone 2 (Blepharismon) and Induction of Pairing

Synthetic gamone 2 was chemically synthesized 3-(2’-formylamino-5’-hydroxybenzoyl) lactate, which is naturally secreted by *B. japonicum* type II cells [20,21,22,23] and was kindly provided by Prof. Hideo Iio and Yoshinosuke Usuki of Osaka Metropolitan University, Japan. Synthetic gamone 2 was dissolved in SMB, filtered, and kept frozen. For the induction of pairing, frozen gamone 2 was thawed and mixed with the appropriate cell suspension. Synthetic gamone 2 (1.6 µg/mL) corresponds to the activity of approx. 500 U/mL in *B. japonicum* strain R1072 (I) and also affects other Mk species [3].

### 2.4. Induction of Mating Pairs between Different Species Pretreated by Gamones

As shown in Figure 1, type I cells were pretreated with synthetic gamone 2 (final concentration: 1.6 µg/mL), and type II cells were pretreated with CFF1 derived from the same species for 3–4 h. When pair formation occurred, the suspension of homotypic pairs was stirred by a micropipette, and the separated cells were mixed with the appropriate cell suspension. Pretreatment of the cells is an indispensable process for synchronizing nuclear changes and creating a timetable.

### 2.5. Judgement of Inter-Species Pair or Intra-Species Pair and Observation of Nuclear Changes during Conjugation

For the assessment of inter-species or intra-species pairs, when the index of pair formation [7] reached >3, pairs of cells were collected directly with a micropipette from the mixture or by mild centrifugation and then fixed and stained as described below. The species of approx. 1500 cells in the mixtures including >100 pairs were classified based on the cell size and macronuclear shape.

For the observation of nuclear changes during conjugation, cells (40–50 pairs) were taken from the aliquot periodically (0, 2, 6, 12, 14, 18, 20, 22, 24, 28, and 30 h after mixing) fixed and stained as described below, and nuclear stages were determined.

The cells were fixed on a glass slide with a mixture of 100% ethanol and acetic acid in the ratio 18:1 or 6:1. After air-drying, the slide was incubated in 70% ethanol overnight to remove the red pigment from the cells. The glass slides were rinsed in distilled water, mounted with a drop of DAPI solution (2.0 μg/mL), and observed under a fluorescent microscope (BX50, Olympus, Tokyo, Japan).

### 2.6. Establishment of Viable Progenies after Conjugation

Conjugating pairs were isolated in SMB, and if the pair was maintained 1 day later, it was designated as a heterotypic pair. Such heterotypic pairs were gradually separated after conjugation, and the exconjugants that survived were isolated in culture medium. If the exconjugants divided at least once, the cells were isolated again in new culture medium as a caryonide. If the caryonide divided and established a clone, and the immature period was observed, it was identified as a progeny.

## 3. Results

### 3.1. Species-Specific Gamone 1 in Three Mk Species

Table 2 shows the pair inducibility of cell-free fluid from type I cells (CFF1) of each species. CFF1 from *B. undulans* (Mk2) only induced pair formation in type II cells of *B. undulans*, and it had no effect on *B. japonicum* (Mk4) or *B. stoltei* (Mk4) as reported [3]. CFF1 from the closely related species *B. japonicum* and *B. stoltei* affected both *B. japonicum* and *B. stoltei*, but it had no effect on *B. undulans*, also confirming previous observations [3]. CFF1 from *B. americanum* (Mk3) did not induce pair formation in *B. japonicum*, as reported [9]. With the use of newly obtained *B. americanum* stocks, we observed that CFF1 from *B. americanum* affected only *B. americanum* (Mk3) and had no effect on *B. undulans* (Mk2), *B. japonicum* (Mk4), or *B. stoltei* (Mk4). Type II cells of *B. americanum* responded to only CFF1 from *B. americanum* and not to CFF1 from *B. undulans*, *B. japonicum*, or *B. stoltei*. Gamone 1 that is present in CFF1 from each species thus appears to be species-specific (or at least Mk-specific) in Mk2, Mk3 and Mk4 *Blepharisma*.

### 3.2. Competence of Pair Formation between Species

Even though gamone 1 is species-specific or Mk-specific, pair formation might occur between species in cases in which both mating-type cells are separately stimulated by their complementary gamones. For example, type I and type II cells of species A and type I and type II cells of species B co-exist and secrete their own gamones. Type I cells of species A secrete species A-specific gamone 1 and stimulate type II cells of species A, and type I cells of species B secrete species B-specific gamone 1 and stimulate type II cells of species B. Stimulated type II cells of both species secrete gamone 2 (which is common among species), stimulating type I cells of both species A and species B. All types of cells are then stimulated, and it is possible that three types of pairs can form: species A–species A, species B–species B, and species A–species B.

For the mating type, homotypic (type I–type I, type II–type II) and heterotypic (type I–type II) pairs are also possible in each combination of species listed above. Homotypic pairs gradually separate, and heterotypic pairs are maintained for > 24 h Our earlier study revealed that even if both mating-type cells of *B. undulans* (Mk2) and *B. japonicum* (Mk4) are stimulated, the percentage of pair formation between species is remarkably low at <2% of total pairs [3]. Interspecific pair formation appears to be rare between *B. undulans* (Mk2) and *B. japonicum* (Mk4). In the present study, we examined interspecific pair formation between *B. undulans* (Mk2) and *B. americanum* (Mk3), between *B. americanum* (Mk3) and *B. japonicum* (Mk4), and between *B. americanum* (Mk3) and *B. stoltei* (Mk4).

#### 3.2.1. Pair Formation between *B. undulans* (Mk2) and *B. americanum* (Mk3)

Type I cells of *B. americanum* (Mk3) were stimulated by synthetic gamone 2, then mixed with *B. undulans* (Mk2) type I and type II cells which had also been stimulated by synthetic gamone 2 and CFF1 of *B. undulans*, respectively. The cells’ pretreatment was as described in the Materials and Methods and Figure 1. One hundred pairs were fixed, stained, and observed by fluorescent microscopy. Species were identified by the macronuclear shape. Sixty-eight pairs were *B. undulans*–*B. undulans* pairs (68.0%), 29 pairs were *B. americanum*–*B. americanum* (29.0%), and three pairs were *B. undulans*–*B. americanum* (3.0%) (Table 3). A similar experiment showed that *B. undulans*–*B. americanum* pair accounted for 5.5%. Pair formation between *B. undulans* and *B. americanum* was thus rare, as in the case between *B. undulans* and *B. japonicum* [3].

#### 3.2.2. Pair Formation between *B. americanum* (Mk3) and *B. japonicum* (Mk4), and between *B. americanum* (Mk3) and *B. stoltei* (Mk4)

Type I cells of *B. americanum* (Mk3) were stimulated by synthetic gamone 2, then mixed with *B. japonicum* (Mk4) type I and type II cells which had also been stimulated by synthetic gamone 2 and CFF1 of *B. japonicum*, respectively. We observed 101 pairs: 29 pairs were *B. americanum*–*B. americanum* pairs (28.7%), 29 were *B. japonicum*–*B. japonicum* (28.7%), and 43 were *B. americanum*–*B. japonicum* (42.6%) (Table 3). The percentage of inter-species pair formation was relatively high between *B. americanum* (Mk3) and *B. japonicum* (Mk4). We also investigated pair formation between *B. americanum* (Mk3) and *B. stoltei* (Mk4): 112 pairs were fixed and stained, and 21 pairs were *B. americanum*–*B. americanum* (18.8%), 16 were *B. stoltei*–*B. stoltei* (14.3%), and 75 were *B. americanum*–*B. stoltei* (67.0%) (Table 3). The percentage of pair formation was thus also high between *B. americanum* (Mk3) and *B. stoltei* (Mk4).

These results indicate that if cells are artificially stimulated, pair formation occurs at relatively high frequency between *B. americanum* and *B. japonicum* and between *B. americanum* and *B. stoltei*, but at low frequency between *B. undulans* and *B. americanum*.

### 3.3. Nuclear Changes in Inter-Species B. americanum–B. stoltei Pairs

In the mixture of *B. americanum* and *B. stoltei* described above, inter-species pairing occurred at a relatively high frequency. We examined whether nuclear changes occur in such inter-species pairs, and when such changes occurred, we sought to determine whether the time course for nuclear changes is the same as in intra-species pairs. The time course of *B. japonicum* pairs has been described [24] (Figure 2). In the present study, we compared the time courses of intra-species *B. americanum* pairs and *B. stoltei* pairs, and inter-species *B. americanum*–*B. stoltei* pairs.

Figure 3 illustrates the time courses for nuclear changes of micronuclei during conjugation in intra-species pairs (*B. americanum*–*B. americanum*) and (*B. stoltei*–*B. stoltei*) and in inter-species pairs (*B. americanum*–*B. stoltei*). Nuclear changes occurred even in the inter-species pairs (Figure 4). In all three combinations, the completion of meiosis took approx. 14 h (although the nuclear changes in *B. americanum* were not synchronized very well); division of the synkaryon took ~18 h, and the formation of macronuclear anlagen took ~24 h (Figure 3). Few differences were thus observed in the time course of nuclear changes between species or between intra- and inter-species. However, the abnormality of nuclear changes was more frequently observed in the inter-species pairs (Figure 5).

Figure 5 depicts the frequency of abnormal conjugation in intra-species (*B. americanum*–*B. americanum*) (Figure 5A) and inter-species (*B. americanum*–*B. stoltei*) (Figure 5B) pairs. Samples were periodically taken from the aliquot 2–30 h after induction. The observed abnormal conjugations were (*i*) exconjugant with an abnormal number (none or >3) of macronuclear anlagen (Figure 5C); (*ii*) prematurely separated mating cells with meiotic micronuclei and without macronuclear anlagen (Figure 5D); and (*iii*) mating cells with an asynchronous progression of nuclear changes (e.g., one of the conjugants showed meiosis stage III, whereas the mate showed stage VIII) (Figure 5E). Such abnormal conjugation was observed more often in inter-species pairs than intra-species pairs (Figure 5).

Although abnormality of conjugation was observed more often in inter-species pairs, most of the pairs underwent nuclear changes such as meiosis, formation of synkaryon, and macronuclear anlagen. Exconjugants, i.e., cells that separated after conjugation, were observed in both intra- and inter-species pairs. The following experiment was conducted to determine whether the exconjugants divide and establish caryonidal clones. We speculated that if this were to occur, the progeny of inter-species conjugation might be produced.

### 3.4. Progeny after Conjugation

Intra- and inter-species pairs were isolated 2–3 h after mixing, and if the pair was maintained after 1 day, it was identified as a heterotypic pair. Such heterotypic pairs were indeed obtained: *B. americanum*–*B. americanum* pairs (*n* = 17) and *B. americanum*–*B. stoltei pairs* (*n* = 27) (Table 4). The pairs gradually separated after conjugation, and the number of surviving exconjugants was 32 (94%) in *B. americanum*–*B. americanum* and 51 (94%) in *B. americanum*–*B. stoltei*. Surviving exconjugants were isolated in culture medium and if the exconjugants divided at least once, the cells were isolated again in new culture medium as a caryonide. Twenty-five exconjugants of *B. americanum*–*B. americanum* (of 26 isolates) divided, whereas no exconjugants of *B. americanum*–*B. stoltei* (of 43 isolates) divided. Most of the exconjugants of *B. americanum*–*B. americanum* established caryonidal clones. The immature period was observed in these caryonidal clones, and these clones were identified as true progenies. These results demonstrate that *B. americanum*–*B. americanum* pairs produced viable progenies, but *B. americanum*–*B. stoltei* pairs produced no progenies at all.

## 4. Discussion

This study revealed that CFF1 from *B. americanum* (Mk3) had no effect on either *B. undulans* (Mk2), *B. japonicum* (Mk4) or *B. stoltei* (Mk4). Type II cells of *B. americanum* responded to only CFF1 from *B. americanum* and not to CFF1 from *B. undulans* (Mk2), *B. japonicum* (Mk4), or *B. stoltei* (Mk4). Gamone 1 (which is contained in CFF1 from each species) therefore appears to be species-specific (or at least Mk-specific). With the inclusion of an Mk3 species (*B. americanum*) to prevent inter-species conjugation, the mating pheromone gamone 1 seems to be the first barrier to conjugation, as we have shown earlier [3].

Pair formation is the second step for preventing inter-species conjugation, as we have shown with the use of *B. undulans* (Mk2) and *B. japonicum* (Mk4) [3]. The present study also confirmed this in *B. undulans* (Mk2) and *B. americanum* (Mk3). Only a few pairs were formed between these two pretreated species by complementary gamones, but we found in the present study that the pair formation occurred at a relatively high frequency between *B. americanum* (Mk3) and *B. japonicum* (Mk4) and between *B. americanum* (Mk3) and *B. stoltei* (Mk4). This study provides the first observation of inter-species *Blepharisma* pairs that underwent nuclear changes specific to conjugation. There were no significant differences in the time courses of the nuclear changes between the intra- and inter-species pairs, but the frequency of abnormal cells in the inter-species pairs was higher than that in the intra-species pairs. Moreover, no exconjugants divided and no progenies were obtained in the inter-species pairs. Taken together, these results indicate that even if inter-species pair formation is successful, there may be another step after pair formation that prevents inter-specific conjugation. The exact timing and the substantial factor responsible for the step remain unknown.

*Euplotes octocarinatus* forms mating pairs with *E. patella* when each species is pretreated with its own mating pheromones [1], but the interspecific conjugation of *Euplotes* was prevented during mostly premeiotic division and meiosis, and all conjugants failed to survive. These findings suggested that (i) the adhesion molecules are not species-specific, and (ii) some factors involved in pre-meiotic division and meiosis are species-specific.

The presumptive steps or barriers of conjugation in *Blepharisma* and *Euplotes* that may prevent interspecific conjugation in both genera are shown in Figure 6 which is a revised one from our previous report [3]. On the basis of our study, we propose the term “step” instead of “barrier” because the mechanisms to prevent inter-species conjugation may be present at certain stage between meiosis of micronuclei and development of macronuclear anlagen in *Blepharisma*, and “step” is more appropriate than “barrier”. Species-specificity of mating pheromones or gamones seems to be the first barrier or step in both genera; however, the second and third barriers or steps differ between *Blepharisma* and *Euplotes* in the conjugation process. Our earlier study suggested at least two barriers, i.e., gamone 1 and specific factors involved in cell adhesion of *Blepharisma* [3]. Our present findings suggest that another step may be present after meiosis (Figure 6). However, these three steps do not always work for all interspecific conjugation. Interspecific conjugation is inhibited in Step 2 and does not progress further between *B. undulans* (Mk2) and *B. japonicum* (Mk4) [3] or between *B. undulans* (Mk2) and *B. americanum* (Mk3) (Table 3). However, interspecific conjugation progressed beyond Step 2 between *B. americanum* (Mk3) and *B. japonicum* (Mk4) and between *B. americanum* (Mk3) and *B. stoltei* (Mk4) (Table 3), and was prevented at Step 3 (Table 4). The details of the differences in steps for inhibition are still unknown, but may be related to the genetic distance between MKs. There are plenty of phylogenetic trees for *Blepharisma* species using SSU rRNA have been reported [25,26,27,28]. However, as the sequence of SSU rRNA is so close from species to species in *Blepharisma* and is not suitable to reveal the relationship between species, other markers should be used for the analysis. Using a mitochondrial gene (cytochrome oxidase subunit 1 gene), our recent study suggests that species of Mk3 are closer to those of Mk4 (data are not shown). It seems likely that the interspecific conjugation between Mk3 and Mk4 progressed beyond step2. The stop at Step 3 may be due in part to the failure of the macronuclear anlagen to develop normally in the interspecific pairs. Regardless of the reason, reproductive isolation in *Blepharisma* appears to be established in at least three steps. Future work will elucidate the mechanism of reproductive isolation within the genus *Blepharisma*.

## Figures and Tables

**Figure 1 microorganisms-11-00188-f001:**
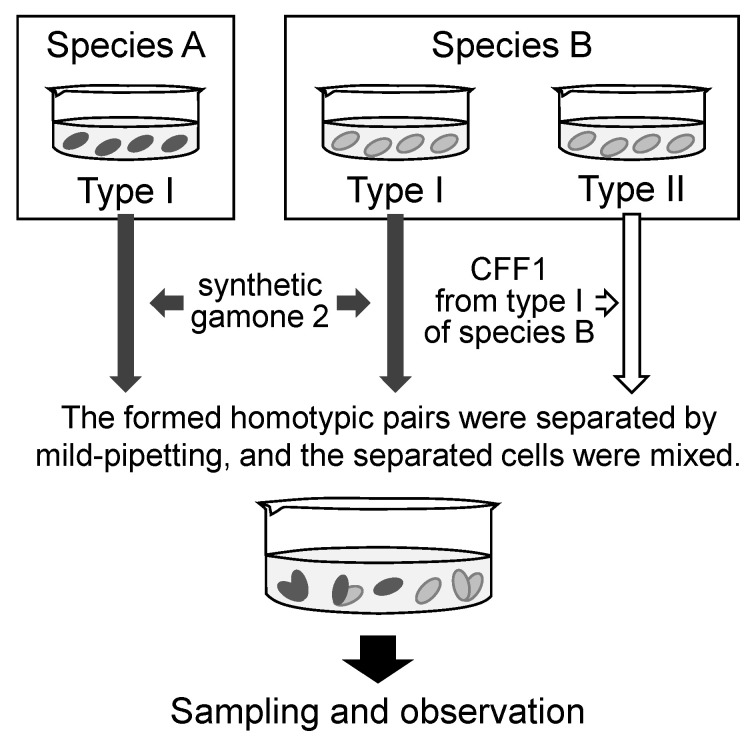
Procedure for the induction of mating pairs between different species.

**Figure 2 microorganisms-11-00188-f002:**
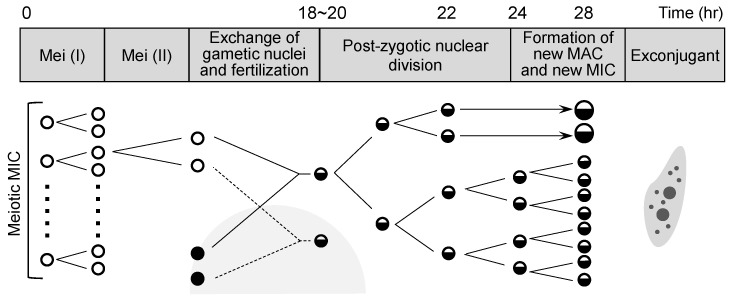
Nuclear changes of micronuclei during conjugation in *B. japonicum* (modified from [24]). Mei (I): Meiosis I, Mei (II): Meiosis II.

**Figure 3 microorganisms-11-00188-f003:**
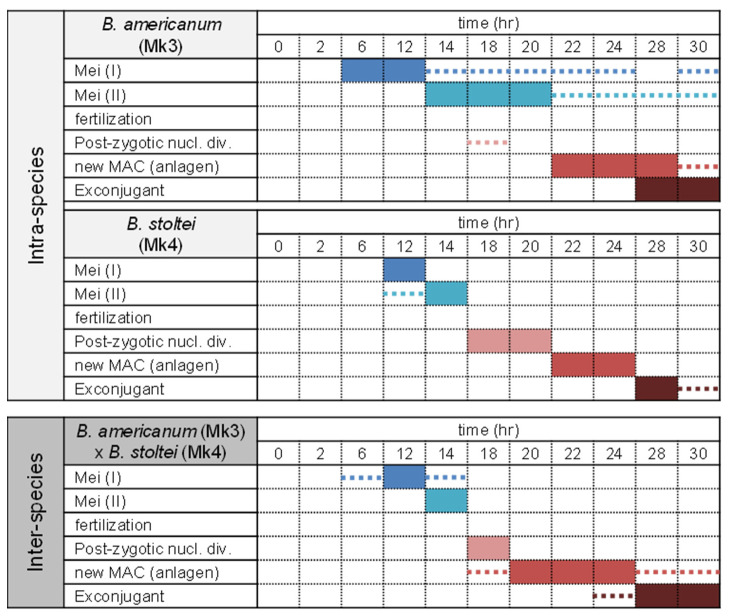
Time course of nuclear changes during conjugation in intra-species (*B. americanum*–*B. americanum* and *B. stoltei*–*B. stoltei*) pairs and in inter-species (*B. americanum*–*B. stoltei*) pairs. Colored boxes: nuclear stages most frequently observed at the indicated time. Dotted lines: stages observed slightly at the indicated time.

**Figure 4 microorganisms-11-00188-f004:**
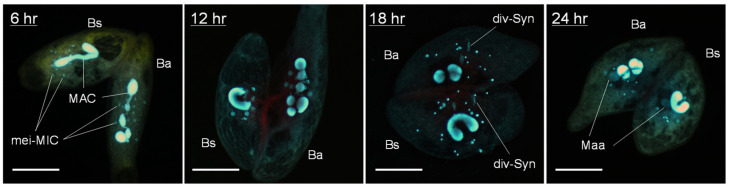
Inter-species (*B. americanum*–*B. stoltei*) conjugating pairs stained with DAPI as described in Materials and Methods. Ba: *B. americanum* (with multinodal macronucleus); Bs: *B. stoltei* (with filiform macronucleus); mei-MIC: meiotic micronuclei; MAC: macronucleus; div-Syn: division of synkaryon; Maa: macronuclear anlagen. Scale: 50 µm.

**Figure 5 microorganisms-11-00188-f005:**
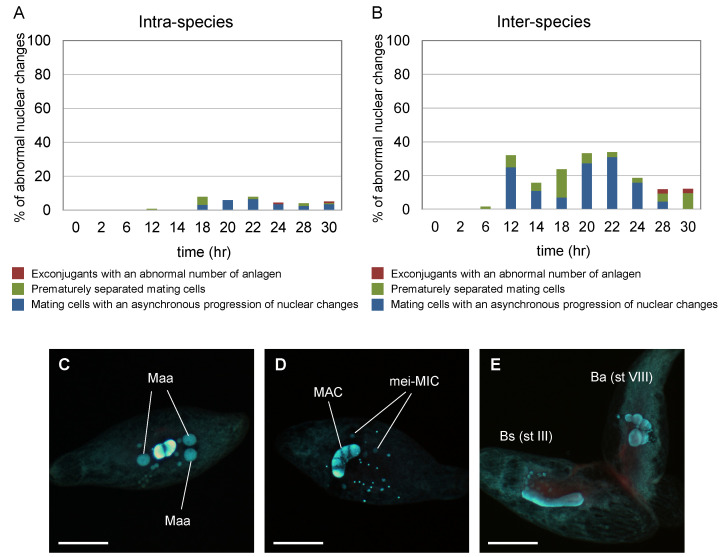
Frequency of abnormal nuclear changes of micronuclei in intra-species (*B. americanum*–*B. americanum*) and inter-species (*B. americanum*–*B. stoltei*) pairs. % of abnormal nuclear changes in the intra-species (**A**) and in the inter-species (**B**). An exconjugant with an abnormal number of anlagen (**C**), prematurely separated mating cell (**D**), and mating cells with an asynchronous progression of nuclear changes (**E**). Maa: macronuclear anlagen; mei-MIC: meiotic micronuclei; MAC: macronucleus; Ba (st VIII): *B. americanum* (stage VIII, metaphase of Meiosis I); Bs (st III): *B. stoltei* (stage III, early prophase of Meiosis I). Scale: 50 µm.

**Figure 6 microorganisms-11-00188-f006:**
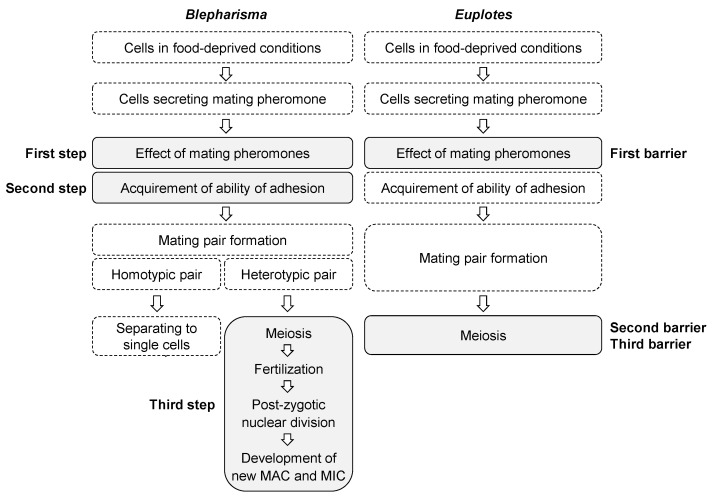
Presumptive steps or barriers that prevent interspecific conjugation in *Blepharisma* and *Euplotes*. This figure has been modified after [3], by the addition of a third step in *Blepharisma*.

**Table 1 microorganisms-11-00188-t001:** Stocks used in this study.

Mk	Species	Mating Type	Strain	Long Diameter	Short Diameter	Collection Site
2	*B. undulans*	I	EN-II	137	60	Ibaraki, Japan
		I	SDT2-II	130	48	Minamidaito Island, Japan
		II	SDT1-II	111	22	Minamidaito Island, Japan
3	*B. americanum*	I	CCAP1607/1	233	105	USA
		II	KR-III	148	44	Gyeongju, South Korea
4	*B. japonicum*	I	R1072	340	164	Bangalore, India
		II	YC-IV	232	104	Iwakuni, Japan
	*B. stoltei*	I	ATCC30299	211	86	Lake Federsee, Germany
		II	HT-IV	199	75	Aichi, Japan

Mk: megakaryotype [18]. Dimensions of cells (long and short diameters) are shown in average (µm).

**Table 2 microorganisms-11-00188-t002:** Species-specific (or at least Mk-specific) pair inducibility of CFF1 from *Blepharisma*.

			Type II Cells			
			Mk2	Mk3	Mk4	
		Species	*B. undulans*	*B. americanum*	*B. japonicum*	*B. stoltei*
**CFF1 of type I cells**	**Mk2**	*B. undulans*	+	**−**	−	−
	**Mk3**	*B. americanum*	**−**	**+**	**−**	**−**
	**Mk4**	*B. japonicum*	−	**−**	+	+
		*B. stoltei*	−	**−**	+	+

Stocks of mating type I cells: SDT2-II (*B. undulans*), CCAP1607/1 (*B. americanum*), R1072 (*B. japonicum*), and ATCC30299 (*B. stoltei*). Stocks of mating type II cells: SDT1-II (*B. undulans*), KR-III (*B. americanum*), YC-IV (*B. japonicum*), and HT-IV (*B. stoltei*). Cell-free fluid from type I cells (CFF1) was mixed with moderately starved type II cells. Pair formation was observed (+) or not observed (−). Bold letters indicate the results newly obtained in this study.

**Table 3 microorganisms-11-00188-t003:** Pairing capacity between species in complementary gamone-pretreated cells ^1^.

	Species Mix		Species Mix	
Mk3	*B. americanum* (I)	Mk3	*B. americanum* (I)	
Mk2	*B. undulans* (I) *B. undulans* (II)	Mk4	*B. japonicum* (I) *B. japonicum* (II)	*B. stoltei* (I) *B. stoltei* (II)
% inter-species pairing	3.0%, 5.5%	% inter-species pairing	42.6%	67.0%

^1^ More than 100 pairs were examined and counted for the percentage of inter-species pairing. Stocks used: CCAP1607/1 (*B. americanum* (I)), EN-II (*B. undulans* (I)), SDT1-II (*B. undulans* (II)), R1072 (*B. japonicum* (I)), YC-IV (*B. japonicum* (II)), ATCC30299 (*B. stoltei* (I)), HT-IV (*B. stoltei* (II)).

**Table 4 microorganisms-11-00188-t004:** Exconjugants that survived after intra- and inter-species conjugation.

	***B. americanum*–*B. americanum***	***B. americanum*–*B. stoltei***
No. of heterotypic pairs isolated	17	27
No. of exconjugants survived	32/34 (94%)	51/54 (94%)
No. of exconjugants divided	25/26 (96%)	0/43 (0%)

Stocks used: intra-species cross, *B. americanum* CCAP1607/4 (mating type I) and KR-III (mating type II), inter-species cross, *B. americanum* CCAP1607/4 (mating type I) and *B. stoltei* HT-IV (mating type II).

## Data Availability

The data of this study are available from the corresponding author upon reasonable request.

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
