# Peer review of "Possible Third Step Preventing Conjugation between Different Species of *Blepharisma"

_microorganisms, 2023, doi:10.3390/microorganisms11010188_

Round 1
Reviewer 1 Report
The reviewed paper dealt with investigation of mechanisms preventing interspecific conjugation in ciliates. It was shown that there are at least three levels of the barriers: mating pheromones or gamones; factors that facts on pre-meiotic division and factors that acts on meiosis. It is quite interesting that that latter mechanism has differences in representatives of genera Blepharisma and Euplotes. The results obtained are quite important for understanding of microevolutionary processes in ciliate populations.
As a remark, it should be noted that the results of the investigation are not quite well formulated in the summary. One gets the impression that the authors investigated only the third level isolation mechanisms, while in fact all three levels have been experimentally studied. In addition, it is not always clear from the text of the article when the authors refer to their previous experimental data, and when to current data.
Author Response
To the Reviewer 1
First, we would like to express our deepest thanks to the reviewer for the valuable comments on our manuscript. After careful consideration of the comments, we respond as follows. We have also revised the manuscript.
The Reviewer pointed out that the results of the investigation are not quite well formulated in the Summary, the authors should state clearly that they experimentally investigated all three levels of isolation mechanisms. Following these comments, in Summary section, we now described that “Using four species, we experimentally investigated other potential barriers to interspecific conjugation in Blepharisma, as well as the first and second barriers.” We also emphasized the present study describing as “These results suggest a third barrier or step, in addition to the first and second ones, in nuclear changes after pair formation that prevents interspecific conjugation in Blepharisma.”
The reviewer also pointed out that current and previous data were not clearly presented separately in the text of the article. We have been carefully checked throughout the article, and described clearly current and previous data in several sentences. In the table 2, the current data is now shown in boldface so that the readers can clearly understand which data is the current one.
We hope that the manuscript has been extensively revised and now ready to publish in the Microorganisms.
Thank you for your considerations.
Reviewer 2 Report
The manuscript is basically very well prepared and describes a novel third step that prevents conjugation between different Blepharisma species. The methodology used is appropriate. The English is fine throughout the manuscript. Thus, I have only minor suggestions that may be considered.
1. Lines 104-107 may be shifted to the M&M rather than the legend of table 1.
2. A plate (photomicrographs) showing images of conjugating pairs may be added.
Author Response
To the Reviewer 2
First, we would like to express our deepest thanks to the reviewer for the valuable comments on our manuscript. After careful consideration of the comments, we respond as follows. We have also revised the manuscript.
Following to the Reviewer’s suggestions, 1) we shifted line 104-107 (table 1) to the Materials and Methods section, and 2) we added a photomicrograph of the interspecific conjugating pairs in the article.
We hope that the manuscript has been extensively revised and now ready to publish in the Microorganisms.
Thank you for your considerations.